# Recruiting Participants in Vulnerable Situations: A Qualitative Evaluation of the Recruitment Process in the EFFICHRONIC Study

**DOI:** 10.3390/ijerph191710765

**Published:** 2022-08-29

**Authors:** Pilar Serrano-Gallardo, Viola Cassetti, An L. D. Boone, Marta María Pisano-González

**Affiliations:** 1Nursing Department, Faculty of Medicine, Universidad Autonoma de Madrid, 28029 Madrid, Spain; 2Instituto de Investigación Sanitaria Puerta de Hierro—Segovia de Arana, Instituto Interuniversitario “Investigación Avanzada Sobre Evaluación de la Ciencia y la Universidad”, 28029 Madrid, Spain; 3Community Activities in Primary Care Programme (PACAP), 18001 Granada, Spain; 4Research Group “Community Health and Active Aging” of the Research Institute of Asturias (IPSA), Consejería de Sanidad del Gobierno Regional de Asturias, Public Health General Directorate, Principality of Asturias, 33005 Oviedo, Spain; 5Research Group “Community Health and Active Aging” of the Research Institute of Asturias (IPSA), Consejería de Sanidad del Gobierno Regional de Asturias, General Directorate of Care, Humanisation and Social Healthcare Services, Principality of Asturias, 33005 Oviedo, Spain

**Keywords:** qualitative evaluation, recruitment, responsible research, stakeholders, vulnerable population, expert patient programme, chronic disease self-management

## Abstract

In recent years, stakeholder involvement in research has become a central element of responsible research. The EFFICHRONIC project reflects these principles and aims to reduce the burden of chronic diseases and increase the sustainability of the healthcare system through the implementation of an evidence-based chronic disease prevention and self-management programme. The qualitative study presented here is part of EFFICHRONIC and aims to explore and understand the recruitment strategies implemented in the participating countries (Spain, UK, Netherlands, Italy, and France). Semi-structured interviews were conducted with the country coordinators (purposive sampling of the five coordinators responsible for the recruitment strategy), and a coding and synthesis process was used to conduct a thematic analysis. The analysis resulted in five main categories: (1) Stakeholder recruitment strategies. (2) Facilitators to recruitment. (3) Barriers to recruitment. (4) Strategies developed to address recruitment challenges. (5) Lessons learned. From a collaborative approach to the co-production process, recruitment has helped to build a wide network and new relationships with local actors, explore and learn about the social world, step out of the comfort zone of health institutions, combine a wide variety of strategies, and innovate by taking into account the institutional and cultural contexts of each country.

## 1. Introduction

Over the past decade, there has been increasing interest in involving stakeholders in research, particularly from a Responsible Research approach. This approach requires understanding how stakeholders can better interact to move toward science that reflects social acceptability, sustainability, and convenience [1,2].

The European Commission [3] has emphasised this participatory approach in research, with the incorporation of stakeholders, who are people or organisations other than researchers who can bring or receive value, whether as beneficiaries, health professionals, policymakers, managers, industry, civil society organisations, NGOs, people with an illness, carers, or citizens. It also encourages using this approach to foster mutual understanding and co-create the results into effective policy agendas to address societal challenges. Along these lines, a variety of programmes have been developed (e.g., NIHR INVOLVE [4]), incorporating these participatory processes as an essential part of the process of identification, prioritisation, design, and dissemination of any research process, given that stakeholders’ involvement impacts the planning of programmes’ actions and the results obtained.

The EU-funded EFFICHONIC project, where the present study is nested and in which Spain, the United Kingdom, the Netherlands, Italy, and France participate, aimed to reduce the burden of the most common chronic diseases and increase the sustainability of the health system by demonstrating the efficiency of evidence-based chronic disease prevention and self-management programmes. It centred specifically on how to reach vulnerable people with chronic conditions and/or their hard-to-reach carers [5,6]. The specific goals of the EFFICHRONIC project, aligned with the concept of Responsible Research, include: identifying vulnerable groups and individuals and designing specific recruitment strategies for vulnerable people and groups, engaging them in the self-management programme, and learning how to manage various aspects related to their health and well-being. These goals are essential to achieve both the general objective stated above and the specific objective “To define policy recommendations and guidelines to enable stratification and adaptation of this chronic disease self-management programme in other regions and countries in Europe” [5].

On the other hand, it is also worth noting the added complexity of the programme implemented at EFFICHRONIC, the “Chronic Disease Self-Management Program” (CDSMP), which originated at Stanford University [6]. The programme, which incorporates peer-to-peer education, aims to increase people’s self-efficacy and ability to make healthier life choices by addressing the physical, relational, and emotional spheres through becoming “expert patients” [7,8]. Through weekly workshops with small groups led by two trained monitors, the aim is to acquire knowledge, skills, and tools to improve quality of life and cope with illness and/or care. The intervention takes place in community settings such as social centres, town council centres, associations, or places of worship. The programme has a fidelity manual that refers to how closely people involved in programme delivery follow the programme as designed, and in this way, increase the quality and homogeneity of its implementation. Its quality standards preserve that, but they have to be adjusted and adapted to each country’s initial situation, previous experience, and resources in the consortium. In fact, as this study will show, in each country, the project had to deal with a variety of different stakeholders, depending on how the local context was organised.

The EFFICHRONIC project thus poses a significant methodological challenge regarding the recruitment of people with chronic health problems and/or their carers with socio-economic or cultural vulnerability, given that this second characteristic especially is not uniquely defined in the literature, and also because the variables that can determine vulnerability at both social and health level are not included in many Health Information Systems [9]. Significantly, in relation to the vulnerability concept, the project included both the clinical and social aspects of vulnerability from the beginning. Clinical vulnerability refers to having a chronic disease or caring for a patient with a chronic disease. As for “social” vulnerability, the definition of the Spanish Red Cross was used as a base: “the zone of social vulnerability is located between the zone of integration (stable work and solid social and family pillars) and the zone of exclusion (lack of work and socio-family isolation). Therefore, this zone is characterised by being more unstable, with precarious jobs, intermittent unemployment, and less-solid socio-family pillars [10]. The final adopted definition reflects those used by the International Red Cross and the World Health Organization. It was agreed upon by all partners involved in the EFFICHRONIC project during one of the initial meetings and defined as: “the diminished capacity of an individual or group to anticipate, cope with, resist and recover from the effect of natural or man-made hazard.” Within the EFFICHRONIC project, the term “vulnerable” included “people suffering from social exclusion and socio-economic hardship, as well as people under physical or psychological stress” [11] (p. 14).

The criteria for inclusion and exclusion of participants (people with chronic diseases and/or their carers) in the EFFICHRONIC project were defined before the start of implementation, with each country making the necessary adaptations according to its social and cultural reality, as well as to the conditions indicated by their Ethics Committees. All of this meant that identifying possible participants to be recruited was highly complex and became a strategic objective in itself, given that it required collaborative work with local stakeholders in each setting as crucial elements in this process.

Defining the theoretical framework of the project, carrying out a stakeholders’ analysis, identifying up-to-date evidence to design actions and proposals, and drafting a base document were common starting points for all pilot studies. In each country, the coordinating team designed the project tasks independently to adapt the project to existing resources and different organisational realities. In fact, EFFICHRONIC’s coordinators had to access the target population from very different professional backgrounds, resources, and organisational structures: university, research, hospital, and governmental institutions. Recruitment had its particularities in each of the five countries in the consortium due to different socio-cultural realities and social and health care models. The project leaders in each country were in charge of tracing this intricate network and are, therefore, the people who know in detail how the process has been carried out.

For all these reasons, carrying out a qualitative research approach with the coordinators of the EFFICHRONIC project involved in attracting and recruiting participants provides enriching and valuable knowledge. The qualitative research results provide added value, not only for the project but also for the scientific community, to enhance the adoption of participatory and cross-sectoral processes to develop the research.

The overall objective of this qualitative study, nested within EFFICHRONIC and coordinated from Asturias (Spain), was to explore and understand the main facilitators and barriers that the coordinators of the recruitment strategy of EFFICHRONIC had in the five member countries.

## 2. Materials and Methods

### 2.1. Study Design and Participants

A qualitative explorative study design was adopted [12]. Purposive sampling of the five coordinators (one per EFFICHRONIC member country: Spain, the United Kingdom, the Netherlands, Italy, and France) who were responsible for the recruitment strategy of participants and monitors for the projects’ implementation stage was carried out. Coordinators’ main duties depended on the specificity of each context and included profiles such as healthcare professionals working in health centres or hospitals, as well as members of social care organisations.

The five interviewees were selected as the key actors who had to connect with all the stakeholders and with the patients themselves, and their perspectives were key to understanding which facilitators and barriers were perceived, thus contributing to obtaining key knowledge that will enhance stakeholders’ involvement in research.

### 2.2. Data Collection

Semi-structured interviews were carried out with the aforementioned key people. The interview, as a qualitative research technique, seeks to explore the subjective experience of people, and, therefore, with this technique, it is possible to obtain in-depth knowledge from people who are experts in the phenomenon under study [13,14].

The interviews were conducted between August and October 2019 via Skype by a researcher with expertise in qualitative research and fluent in English, Spanish, and Italian to ensure the best communication with the interviewees. Each interview lasted approximately 60 min, and all were recorded and subsequently transcribed as specified in the consent form. A question guideline was used to conduct the interview (see Box 1).

Box 1Questions guideline to conduct the interviewCould you start by telling me how was the process of recruitment of stakeholders in your context…How were all relevant stakeholders proposed/found for the recruitment work?
oWhom did you contact?oHow did it go?oor …Which strategies have been most effective in identifying and involving these stakeholders? And the difficulties?Which strategies have been most effective in recruiting vulnerable people (patients and caregivers)?Based on your experience, or according to what stakeholders have said: Have there been difficulties in recruiting users? What could have made it challenging?Have the tools and procedures designed to recruit them been useful? What could be im-proved? What would you change?What have you learnt from this experience of collaborating with local stakeholders for the recruitment of vulnerable populations?How has your relationship with the stakeholders changed after engaging in this process?How would you think you would do the recruitment process again after this experience?Would you like to add something about the topic that was not covered in the interview?

### 2.3. Data Analysis

A thematic analysis was carried out with a coding and synthesis process, assigning codes to one or more words in the narrative to identify different meanings. In the first reading, codes were attributed to small parts of the text. Then, similar codes were grouped into more general themes that continued to reflect the original meaning identified in the texts but at the same time aimed to represent a more abstract concept and could therefore be transferred to other contexts [15,16]. NVivo software was used for the analysis.

To enhance the credibility of the study, two researchers initially coded two of the transcripts separately and then compared the identified codes. This supported the main researcher to check whether important aspects were left out of her analysis because of unanticipated bias on the topic. The two researchers agreed on the identified codes, and they were then discussed together with the rest of the team, who had also read the full transcripts. Moreover, in writing the results section, quotes from transcripts were used to support the description of the findings [17].

### 2.4. Ethical Considerations

The Ethics Committee for Research with Medicines of the Principality of Asturias (no. 118/19, 11 April 2019) approved the present qualitative study nested within the EFFICHRONIC project, which the corresponding Ethics Committees approved in the countries where it was necessary. Written informed consent from the participants was obtained, and the confidentiality of data was guaranteed. The directives of the Helsinki Declaration regarding the ethical principles for research involving human subjects were followed.

## 3. Results

The thematic analysis of the interviews resulted in five categories: 1. Stakeholder recruitment strategies; 2. Facilitators to recruitment, with the subcategories of strategies for monitoring recruitment and participant recruitment; 3. Barriers to recruitment, with the subcategories of difficulties in recruiting monitors and recruiting participants; 4. Strategies developed to overcome the identified barriers; and 5. Lessons learned (Figure 1).

### 3.1. Stakeholder Recruitment Strategies

Due to the variability of possible stakeholder profiles and the different contexts of the European consortium members, recruitment strategies varied from country to country.

Various strategies, such as workshops, conferences, and meetings, were used. Contact with different institutions in the health service structure (especially those related to the project site) was a strategy used in the consortium countries, except for one of the countries that outsourced this process to other organisations already running CDSMP peer-education programmes. How the message was delivered to the institutions was also a key issue in involving the different stakeholders.

“We contacted many patient associations for the first time. … political institutions at the level of the region, or department, or the ministry of public health … we started to meet with them individually and we talked to them about the project” (P1).

“That we go and present it somewhere and they see how enthusiastic we are, even though it is a presentation that I strongly believe in it and that this can work. … and this push can be given by the doctor or the nurses … and you go because someone told you well. However, it does have to come out of emotion” (P2).

“We agreed what vulnerability looked like in [name of country] and how they would approach vulnerable people, and how they were getting them involved in the programme. So it was very much done with them. Not just us telling them, it was done with them. A co-production we called it” (P3).

More social associations and organisations, as well as municipal structures, were also contacted.

“There are the councillors who contact us and act as intermediaries with the associations. Furthermore, if you act as a liaison, it is a different story because the municipality also finances the associations and therefore, a more potent synergy is created. The municipality acts as a spokesperson, the associations respond more easily, and doors open more easily. You can contact, you can organise things more easily” (P4).

“I was searching for organisations like this, through internet and my network, and I had a list of 3 organisations like this. Additionally, this one happened to be around the corner from my home. So I just contacted them” (P5).

### 3.2. Facilitators to Recruitment

Working in co-production made it easier to launch the project, taking advantage of the parties’ expertise. The dissemination of the project at every opportunity (such as at meetings or events) also contributed to enhancing recruitment and strengthening the network.

“it was essential [the co-production], because they are the experts, not us, we are the experts in the quality, they are the experts in the [Stanford COURSE], so yes, we developed it with them, then with the concept agreed, it became the criteria for the project” (P3).

“the study’s Principal Investigator spoke about the project at many important institutional events, so it was taken to various policy and working fields, even at the regional level” (P4).

Moreover, the role of stakeholders (such as associations, organisations, or institutions) was found to be highly relevant. Their social recognition (based on religious or socio-cultural values) enhances credibility and leadership and contributes to facilitating trust and collaboration from potential participants. Especially patient associations, which promote their members’ empowerment, play a clear leadership role in recruitment. Health systems were the key entry points where the programme’s implementation started (except for two countries), and their professionals were central to its success since, if they perceived that the programme worked, they would collaborate.

“Muslim animator, (…), also works for the (health institution) because she is a nurse, and the Health Region told her to do it” (P1).

“Associations like [name of association], for example, a bit linked to the church, and they have a fairly loyal audience there” (P2).

“health staff who already dealt with chronic conditions and therefore worked in laboratories where there were caregivers, (…), so they knew the reality and were purely self-motivated, even personal interest, that some, being chronic or caregivers of parents with major chronic diseases, wanted to be involved” (P4).

Additionally, the rural setting proved to be particularly favourable, given the scarcity of community programmes or interventions and the existing medical desertification. There is less competition between activities than there may be in urban areas, so any action is very welcome. Additionally, engaging people with specific health problems and limited healthcare services acted as a trigger for participating in the programme.

“In rural areas, there is not so much, and so in rural areas, where we are, it is much easier because we don’t have to fight or compete …” (P2).

“The people who came to me are the ones with fibromyalgia because they don’t know anything and don’t know where to go” (P1).

#### 3.2.1. Monitor Recruitment Strategies

Technologies such as the intranet via screensavers, or social media, such as a Facebook page or videos on a hospital’s YouTube channel, were used in some cases to attract and recruit monitors. The snowball strategy was also used to recruit monitors.

“for people at the hospital we just recruited through the Intranet, or facebook, we have an internal facebook page, we have screensavers …” (P5).

“That girl who is a judge, for example, is the sister of another monitor, (…) and she said “well, I’m interested in that too” (…) It’s been a bit like that, like a snowball. And it has been like that… well, the training is open to health workers, but on the other hand, it is open to different types of people” (P2).

In one country, the hospital’s Human Resources department was a strategic point for locating potential monitors, and even seniority or merit criteria were used because a lot of people were initially interested. Interviews were completed to look for suitable profiles, and in general, the candidates were motivated and interested in the project’s social impact. Incentives were also linked to the recruitment of monitors, especially in the health sector.

“I conducted interviews with the participants [monitors] who I thought were more communicative and willing to do courses, and [selected them also] based on the personality characteristics that emerged in the first course” (P4).

“A nurse who conducts a workshop with me now does it during her working hours. And when the workshop is after hours, she gets compensated with time off work” (P2).

Extending recruitment to non-health professionals, with an emphasis on commitment and volunteerism, was also implemented as a recruiting strategy.

“(…) among them there are people from prisons, social workers from day centres, whom we had not contacted, and there are, for example, people from NGOs that we went to introduce them to the programme, and they said, “oh well, I’d like to do the training”. Then there are people … like a judge, a retired doctor, in other words, a group of committed people, who want to do this in their free time” (P2).

#### 3.2.2. Participant Recruitment Strategies

A variety of strategies were used to recruit participants: for instance, engaging influential health professionals from Primary Health Care and social workers (they are identified as extremely valuable figures, very involved and aware of the project); reaching out to the tertiary sector (voluntary work, associations); announcing recruitment through newspaper advertisements that were distributed door-to-door or through letterboxes; through specific newspapers (e.g., for over 55 s, with dissemination in public spaces such as libraries, supermarkets, swimming pools, social work organisations), placing posters in the waiting rooms of health centres; creating an institutional web page where people could register; and finally by publicising it through radio interviews.

“one way to recruit citizens was also through our hospital, through the intranet…” (P5).

“The doctor or the nurses can give this push, and they say, “Hey, I think this could be good for you”, and they push them, and you go because someone told you so” (P2).

“lots of their recruitment is done through the third sector, voluntary sector, through their own communities, not through the health route” (P3).

“Also we did advertising in a [city] newspaper, in 2 different moments: one that goes door-to-door, they put it in the mail boxes, and we did it there twice, once in the summer break, it was successful” (P5).

“so we did another advertisement in the same paper in September but responses are still coming in, so I don’t know, I don’t think it will be as successful as during the summer, so now we will try another newspaper, that is specifically for age 55+, but this is spread through public places […] like in libraries, social work organisation, swimming pool, supermarkets. And I had never heard about it, and since I [‘ve] heard about it, I see it everywhere” (P5).

Associations were also asked to promote the project in their meetings; to give the information to their relatives, friends, and course participants; to take leaflets to their family doctors; to involve them. This was also promoted to employees of the hospital and the municipality on the intranet.

“we continued organising, sending newsletters, initiatives to associations, inviting the population” (P4).

“It could be anyone [employee from the municipality], we recruited really broad. It was on their intranet, it was really the same, because now we are only talking about the train the trainers, but at the same time we were also recruiting citizens” (P5).

Moreover, the monitors themselves recruited participants, and even the participants themselves contributed to recruiting future participants. Word-of-mouth also appeared to function as a mechanism for recruiting participants.

“The one with diabetes, the presenter, was a Muslim, and that’s why they came because they were his networks” (P1)

“It’s the most direct word of mouth that works. So, either from a health professional or from a friend or acquaintance, who explains to them what it’s all about” (P4).

Working collaboratively with those in charge of recruitment and who had already detected the vulnerable population was also a strategy adopted.

“We wanted to see where the most vulnerable population resides [in the region], and a mapping was done, so great, and we started with the most vulnerable areas” (P2).

“we agreed what vulnerability looked like in [name of country] and how they would approach vulnerable people, and how they were getting them involved in the programme. So it was very much done with them. Not just us telling them, it was done with them. A co-production we called it” (P3).

### 3.3. Barriers to Recruitment

The required processes to comply with ethical requirements slowed down project implementation in some countries, but cultural issues also acted as barriers.

“We started much later with the courses because, as a centre (in a country), as a hospital, we need the approval of the ethics committee” (P4).

“to talk about an ethnic minority, in (a country) is forbidden by the constitution” (P1).

“[what] we haven’t done is go for [implementing the programme in] prisons. Be cause ethically [it] would be practically impossible to do in (a country)” (P3).

Furthermore, a lack of coordination between the healthcare and the welfare sectors emerged.

“There are two types of associations or groups, either social or health, it is impossible to find one that does both. So when you go to a health association, like a patients’ association, they don’t have vulnerable people, because whoever does this action, to join a patients’ association, is not vulnerable” (P1).

In one of the countries, it was pointed out that the CDSMP programme was already seen as something old and that the institutions/politicians no longer support these types of programmes but rather bet on those based on digital technology, and, therefore, funding, and more so in times of cuts, is directed towards these new programmes.

“people are moving on to other things, like into more digital based programmes as opposed to group-based programmes. (…) it’s run locally by mainly voluntary sector organisations, struggling for funding to be honest at the moment, and the country funding are being cut” (P3).

The project impact analysis requirements were seen as a constraint, with a lot of red tape involved. Filling in additional questionnaires on top of those already involved in the programme was perceived as challenging.

“getting people to fill in the questionnaire is … Is proving quite hard. One of the things we realised of course is that people already complete questionnaires for the programme providers [who organise the courses and recruit participants]” (P3).

On the other hand, there were more logistical problems, such as the limited implementation time of the project, which had drawbacks for recruiting and training monitors, as well as hardly any time to conduct the workshops with the participants. Furthermore, problems of accessibility to the places where the workshops were held and insufficient human resources to carry out the training were also mentioned.

“At 14.30, the older people are asleep … and so … yes, we finished it, but with great difficulty, we started with 10, and we didn’t even have 5 [at the end]. It was really exhausting” (P4).

Inadequate awareness of the CDSMP programme, or of the project itself at the outset, together with the challenge of not having to “compete” with other pre-existing programmes, also emerged in some interviews.

“In the first wave for the animators [monitors], we were actually very selective because we were afraid we would not find them, because in other countries there are no such programmes, but here, even if you talk about self-management of the disease… it is the basis of all the programmes, there are so many” (P1).

#### 3.3.1. Difficulties in Recruiting Monitors

In one country, it was reflected that if remuneration was not offered, it was more difficult to recruit potential monitors.

“The first problem that the animators [monitors] wanted to get paid” (P1).

The timing of the recruitment also appeared to be important, as it is a barrier if too much time elapses between recruitment and the start of workshops for participants.

“Many people said that they could no longer take part. Some because of illness problems, others because of important family situations that had arisen in those months” (P4).

#### 3.3.2. Difficulties in Recruiting Participants

The very vulnerability of the potential participants (those who are older adults, have mobility problems, or live/are alone), their culture (they do not want to leave home, they are suspicious of what is free, older adults), their migratory situation (not having a regularised administrative situation, stability in the host country, language barriers for immigrants, etc.), were also seen as obstacles to recruitment.

“It’s very old people, people who live alone, and frankly distrustful, so you called them, and they just didn’t trust that it was that good “you’re really going to give me a workshop, for free…” (P2).

“They [migrants] are afraid that the police come while we give workshops” (P2).

“Because they are the people [in marginalised situations] who are less involved in the advocacy and outreach meetings that the hospital offers. Furthermore, they are the people who have the most problems, even if they live in situations of isolation. So then how do we reach them? (P4).

### 3.4. Strategies Developed to Address Recruitment Challenges

Combining strategies was essential for recruitment. Innovating and being creative in exploring avenues (new partnerships, contacts, committed people) worked well, along with knowing the institutional and socio-cultural idiosyncrasies.

“I think it has to be a combination, because any strategy is difficult, so we need a combination in order to get numbers [reaching enough participants]. But I hope, now we started flyering, so we have our hopes up again, but we know from experience” (P5).

“In urban areas, thank God or thanks to the fact that there is a history, we now have an associative fabric and a community fabric in which you enter, for example, a social centre and you have Tupperware cookery classes, patchwork classes, belly dancing classes, classes of whatever you want, they have, so to speak, a wide range of ways of spending the afternoon” (P2).

A significant amount of recruitment was made from the third sector (volunteers). Working with small associations and organisations engaged with local communities was an unanticipated but highly valued strategy. These associations easily see the value of the programme and deal directly with the programme deliverer. Associations respond more readily and doors open more easily when the municipality acts as a spokesperson.

“For me, it’s very difficult to do the intake. For the volunteer agency, they know the people, also, so it’s easier for them” (P5).

“Now it is easier, under the patronage of the municipality of (name of the city), because we are stronger, sponsored by the municipality of (name of the city), (…). So, regarding citizens, we can reach more people” (P4).

Involving health professionals, especially in hospital settings, was a tactic to address recruitment difficulties. Increasing the number of staff in the project, as well as teamwork and articulating considerations, such as giving the staff days off, made it possible to overcome the obstacles perceived in the initial stages.

“The first thing we did was to get there and have them take off their white coats, we literally took off their white coat and sat them down to shake them a little bit from the mega interventionism that you have in a hospital and shake them up, so that they would think a little bit like normal people, and like someone who has a mother and a father” (P2).

“We had another learning… in our team group, now we have 3 people working for effichronic on recruitment and also to keep in contact with the trainers, and for me this is also a good learning experience, to guide them [the new co-workers]” (P5).

### 3.5. Lessons Learned

The people interviewed said that it is essential to know the programme better from the beginning to be able to explain it to stakeholders, as well as to be involved in the project from its design, to facilitate engagement. It is seen as essential to involve associations and their managers from the very beginning.

“Until you attend an EFFICHRONIC course, you don’t know what is actually happening” (P4).

“In an ideal world, I’d have been involved from the beginning and work it all through. We really had to move very quickly to get people agree. So if I had to do it again, I would start it earlier” (P3).

“I was frightened about the commitment, in the sense that … I mean … joining the project also asked them for a commitment to promote and engage people. So, maybe some associations, rather than sending the message that this project exists, have not been very active in recruiting [participants] among their members” (P4).

On the other hand, the programme must be credible for all stakeholders and must not just be seen as research but also as a personal learning process to improve one’s own health and, from there, as something that can be established within the health system. It was highlighted that it is important to become involved in the project by experiencing it first-hand.

“And with the participants, the jewel in the crown, so to speak, you are not paid anything. But you get the satisfaction of how people change and how they pick it up, and that’s good” (P2).

“I appreciate personally that I came to learn about this world that is so close to my world, you don’t see it if you are not in it, or working in it. So for me that was really nice, and also [I learnt that] you need another approach, you need to approach people differently …” (P5).

Trusting the programme’s methods, having experienced monitors (such as former course participants, as the core of the programme is learning from other people who have a chronic illness), and maintaining close contact with the research team were all reported as key to the implementation of the project.

“Last year we conducted a lot of workshops, and the more I ran, the more I got out of the way to let them act. Because it works …you see how well they [the sessions of the programme] are constructed …and it works because it is well built…” (P2)

“We think it’s the heart of the programme, learning from others who have a long term condition…” (P3).

As for recruitment strategies, it is important to accept a certain degree of uncertainty as to whether they will work or not. Interviewees reported that it would be more effective to start the courses rather quickly after the recruitment of monitors and participants to avoid the risk they would lose interest should they wait too long to begin the training. In addition to timing, multiple strategies should be adopted from the beginning, such as hanging posters, newspaper advertisements, and flyers, and sufficient time needs to be dedicated to the whole recruitment process.

“We trained them in November, but the courses started almost a year later, in September, because apart from those pilot courses in May, then between the ethics committee, but also the project itself with the leader [monitor], the partners, they all left later, in 2018 and, therefore, in a year and more, situations change” (P4).

“really aggressive in the first week, so poster, newspaper, flyers, so you have a lot at once” (P5).

Logistical issues (timing and location of courses or seasonal considerations) are identified as factors that need to be handled flexibly to ensure a more successful implementation of the programme.

“Both the directors and the association tell us that at the moment [if participants have to reach distant areas of the city, because it is hot] there would be nobody” (P4).

Giving financial recognition to monitors is a proposed improvement strategy, as they are a motivational lever, either in the form of compensation in days off for those working in the health system or financial remuneration for those who are not health workers.

“I would establish a budget for the directors [EFFICHRONIC coordinators], because even though … we did it voluntarily, (…). in reality, it is a big commitment, so even recognition for the directors is a motivational lever” (P4).

“(…) that time they put in would be reverted to days off work. Right now, we are looking at people who don’t have a job in the health system because they are patients or so on, so they can be given money, we don’t know how much yet” (P5).

## 4. Discussion

The EFFICHRONIC project requires a truly cross-sectoral spirit to reach vulnerable populations. Stakeholders’ involvement was not considered from the outset in all countries, but it was ongoing throughout the project’s implementation because it was seen as a necessity and found to be effective. As Evans et al. [18] point out, the motivations and actions of stakeholders, especially those who provide services, are vital to understanding the mechanisms of a programme, and from there, they can play a crucial role in making it work, because if they are involved, the effectiveness of the programme increases. Participatory research helps to ensure the success of the research both culturally and logistically. It has many advantages, including improving the recruitment capacity; generating competition in stakeholder groups; generating productive conflicts that follow from good negotiations; increasing the quality of outputs and expected outcomes; increasing the sustainability of intended objectives beyond funded timeframes and during external funding gaps; and creating system change and new, unanticipated projects [19].

Multiple strategies have been developed for recruitment, especially of participants, in a combined and collaborative approach: information and communication technologies (ICTs) (websites, advertisements, newspapers, posters, radio, etc.), influential health professionals, highly committed social workers, the tertiary sector, snowball and word-of-mouth mechanisms, remuneration/compensation actions, and mapping of vulnerable populations. Some of these strategies were also used to attract future monitors. The use of digital tools for participant recruitment and retention has been explored in the field of clinical trials; social media, websites, e-mail, and TV/radio have been the most studied, but knowledge about their effectiveness is still an open question [20]. Hughes-Morley et al. [21] found that advertising did not improve participation in a trial to determine the impact of advertising focused on patient and public involvement in research on patient recruitment. However, the authors argue that it could have an alternative positive impact by making studies more attractive, acceptable, and patient-centred.

Effective strategies depended on the country, its resources, and the training programme’s timing at the beginning of the project. One fact to consider is that the different teams’ coordinators joined at various points in the project, and not all of them from the beginning. Although this variability could be seen as a limitation, the implementation of strategies with existing resources and from different starting points meant that adaptations were made (e.g., in the Netherlands and Spain, extra staff joined the project) as required by each context, and this is a strength in itself. Although recruitment strategies such as snowballing or word-of-mouth were initially identified, it should be noted that, as a common element for all countries, there was a learning process since it was a novel project in which new strategies had to be explored, and a priori, it was not explicitly known how they would turn out. On the other hand, some strategies that did not work at the beginning bore fruit as the process developed. Flexibility about how different actors would be involved and establishing ongoing contact throughout the research process are seen as keys to sustaining engagement [22].

Working intersectionally, extensive dissemination of the project, working with stakeholders with social recognition to ensure credibility and leadership, and carrying out the project in rural settings have all been commonly identified as facilitators for recruitment. This reflects what Burton et al. [22] point out through a review of Patient and Public Involvement, in that effective co-production is underpinned by good communication and the building of relationships and trust between researchers and patient and stakeholder representatives.

On the other hand, there is an underlying ethical issue that is certainly central, particularly in this project. Vulnerable populations have historically been left out of research precisely because of their vulnerability, and although they are recently becoming the subject of study, as in the case of EFFICHRONIC, they still seem to be a challenge for research ethics committees. The ethics committees’ requirements have been very decisive obstacles in getting the project off the ground in two of the settings, with a delay in the start of the recruitment strategy as a repercussion, specifically in one country, with the impediment of access to the prison population. Adams et al. [23], in a qualitative study on staff responsible for recruitment and retention of clinical research participants, revealed that the conventional approaches of national ethics committees were seen as barriers to the incentivisation or adequate compensation of patients in research.

Moreover, different countries have different welfare and healthcare systems, and social and cultural differences have inevitable idiosyncrasies that create different problems and barriers to recruitment. In this respect, it should be noted that there were very country-specific strategies, such as using the hospital’s human resources department, which cannot be generalised. As Jagosh et al. [19] point out, the efficacy of the implemented interventions is context-dependent, and success is a function of how mechanisms are articulated for each setting.

Other barriers to recruitment identified in the present study were coordination gaps between healthcare and social sectors, rigid institutional hierarchy, and bureaucracy that can make the recruitment protocol inefficient. Additionally, both the limited time and insufficient human resources to implement the project and the competing risk that potential participants could be enrolled in other studies proved to be significant barriers to the project. Moreover, many of the healthcare workers who could recruit patients had too high of workloads to be able to do so.

Several barriers were furthermore labelled regarding the recruitment of monitors. Firstly, the fact that remuneration was not established beforehand; secondly, the non-recognition of the programme’s value; and thirdly, the weak institutional or political support to the programme due to budget cuts or higher interest in technology-based programmes.

More logistical issues were also highlighted, such as problems of accessibility to the place where the workshops were held and the synchronisation between monitors’ training and the implementation of the intervention. The very vulnerability of the potential participants (older people, loneliness, migration status) proved to be an obstacle to recruitment. Adams et al. [23] identified similar barriers, which they classified into three categories: competition from research participants at the organisational and national levels, tension between research workload and clinical workload, and imbalance between personal patient burden and potential benefit. Similarly, in their study, Harrington et al. [24] discussed the limited experience and training of recruiters, their time commitment, and the vulnerability of participants (pregnant women, children and older people, especially those aged 85+) as barriers.

Lessons learned include the importance of understanding the programme in depth from the beginning on the one hand, as well as knowing the associations (stakeholders) and resources of the local context to create networks on the other; working on the credibility of the programme with all stakeholders, from experience, trusting the methods as a learning process; recruiting monitors and participants at the same time; involving strategic professionals, such as primary care or community health professionals, in recruitment; considering the logistical issues required for each context; broadening the target to a younger population; developing incentive systems (remuneration/compensation); and having more time to implement such projects. This reflects the results of a realist evaluation [18], which demonstrated how stakeholders’ pre-existing beliefs about the success of a programme could shape their perceptions of credibility, suggesting the need to manage stakeholder engagement at all stages. Similarly, Berk et al. [25] note that having a Stakeholder Recruitment Committee in place at the start of the study is critical in developing a comprehensive recruitment strategy, which should involve a multi-modal approach, adapting the protocol to reduce complexity, and analysing vulnerable participants to design access to them appropriately.

Finally, it should be noted that the results of this research are in line with the work of Kramer et al. [26], who point to four domains of recruitment challenge as lessons learned: (1) the availability of participants’ time and that they really felt that what they were being recruited for was a problem to be tackled; (2) at the institutional level, financial and staff hiring incentives; (3) the recruitment team needs continuous training; and (4) the design of the intervention has to be attractive to participants and not have any drawbacks. These authors propose that participants and stakeholders become involved in study design and problem-solving, with flexibility in data collection, and that the intervention should meet patients’ needs.

### Study Strengths and Limitations

The main limitation of this study is the limited number of participants interviewed. Nonetheless, this was due to the specificity of the way in which the EFFICHRONIC project was carried out and the precise topic on which this nested study was centred (perceived barriers and facilitators to recruitment). Nonetheless, it should be noted that the variability of the profiles of the interviewees and of the contexts of implementation allowed for a cross-case comparison [17] and provided novel insights into the recruitment process at an international scale, which has rarely been studied. It is expected that despite the small number of participants, the present analysis still provides interesting findings on the barriers faced in recruitment and useful strategies that can be adapted to other settings.

## 5. Conclusions

To conclude, the recruitment process was a novel experience for all countries, as it enabled creating an extensive network and new relationships with local agents, exploring and getting to know the social world in-depth, leaving their comfort zone (health institutions), combining a wide variety of strategies, innovating with creativity, and taking into account the institutional and cultural idiosyncrasies of each country, and all from a collaborative approach of co-production. Along the lines of this paper, future research, as also noted by Burton et al. [22], should aim to explore the experiences of patient and stakeholder participation and engagement in the research process, assessing its strengths and weaknesses from all perspectives.

## Figures and Tables

**Figure 1 ijerph-19-10765-f001:**
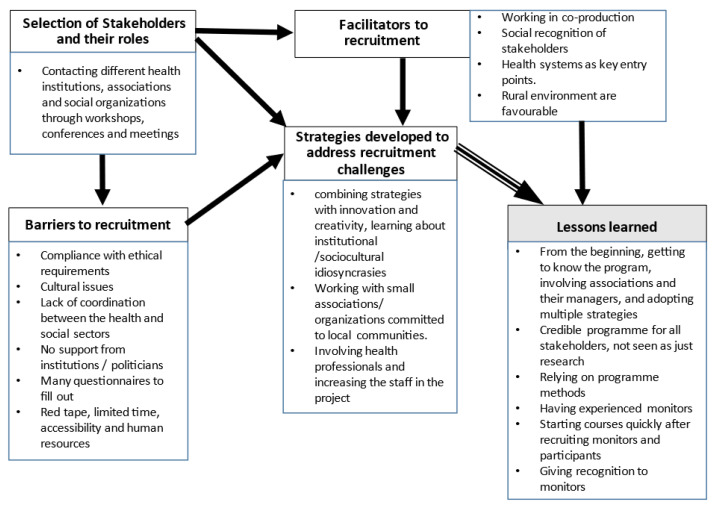
Categories from thematic analysis.

## Data Availability

Not applicable.

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
