# Peer review of "Recruiting Participants in Vulnerable Situations: A Qualitative Evaluation of the Recruitment Process in the EFFICHRONIC Study"

_ijerph, 2022, doi:10.3390/ijerph191710765_

Round 1

Reviewer 1 Report

Dear authors, 

This paper is very well-presented and has demonstrated new knowledge in this field. The discussion and diagram provided lots of insight into issues faced by coordinators for the EFFICHRONIC project. I have only a few suggestions for the authors to enhance this manuscript: 

1) Line 490: "3.6 Figures..." seems to be out of the place in this part of the manuscript. Can the authors consider shifting Annex 1 under the "Method" section instead?

2) Can the authors provided some information about how they ensure "trustworthiness" of their study findings, e.g. how can the readers be assured that the themes were generated in a credible and dependable manner?

Author Response

Dear authors, 

This paper is very well-presented and has demonstrated new knowledge in this field. The discussion and diagram provided lots of insight into issues faced by coordinators for the EFFICHRONIC project. I have only a few suggestions for the authors to enhance this manuscript: 

Thank you for your feedback, please find the corrections below and in the manuscript.

  • Line 490: "3.6 Figures..." seems to be out of the place in this part of the manuscript. Can the authors consider shifting Annex 1 under the "Method" section instead?

Thank you for the suggestion, we moved annex 1 in the methods section as Table 1, and removed the 3.6 subtitle.

  • Can the authors provided some information about how they ensure "trustworthiness" of their study findings, e.g. how can the readers be assured that the themes were generated in a credible and dependable manner?

We added a paragraph to describe with more details the analysis process as a collaborative process within the research team, we hope this helps the reader to understand the credibility and trustworthiness of the findings. We added a reference to cite, on issues of rigor in qualitative research

To enhance the credibility of the study, two researchers initially coded two of the transcripts separately and then compared the identified codes. This supported the main researcher to check whether important aspects were left out from her analysis because of unanticipated bias on the topic. The two researchers agreed on the identified codes and were then discussed together with the rest of the team who had also read the full transcripts. Moreover, in writing the results section, quotes from transcripts were used to support the description of the findings [17].

Reviewer 2 Report

Firstly congratulations to the authors for their work. I hope that my methodological comments will help you give more rigor to your study. 

Abstract

Being a qualitative study, I suggest changing the verbs “identify y analyze” as they are verbs typically used in quantitative studies. Think of verbs like explore, investigate or others.

Introduction.

The introduction is concrete and clear. I suggest the possibility of incorporating more bibliography in relation to, for example the "Chronic Disease Self-Management Program" (CDSMP) (pag 2, line 60-73) and the concepts of vulnerability (pag 2). The authors suggest a definition, but was this the adapted one for all countries, or just Spain? It may be nice to know how these criteria were agreed upon between countries.

In this section, the entire EFFICHRONIC program must be presented, not in methodology. You will see that later on I suggest removing the section on Patient and public involvement.

Material and Methods

Separate the section into two: on the one hand design, and on the other the participants and context (pag. 3 line 116).

1)    Study design: A qualitative design is presented but the type needs to be specified. I suggest the authors look for methodological authors at the level of descriptive qualitative designs. And I suggest you read this article: doi:  10.1177/1744987119880234

2)    Participants and context: The truth is that the study is very interesting. When interviewing only 5 coordinators (these are the participants of this research) it is an important limitation (include in the limitations section). For this reason, it is essential in this section (or in results) to describe the study participants (coordinators). I suggest making a table (age, sex, basic training, advanced training, ..). Also describe the function of the coordinators within the program. It is important for the reader to be able to identify the profile and functions of the coordinators in order to understand the results. In this way it is better understood and it justifies why the study is carried out with so few participants.

It would be necessary to briefly define the context in which the program has been carried out in the 5 countries.

3)    I do not fully understand the Patient and public involvement section (pag. 2, line 121); personally I would remove it. The first part (line 122-126) is included in the introduction and the second part (line 16-130) details the characteristics of the participants. I would include it in the Participants and context section. That is, the EFFICHRONIC program is presented in the introduction.

4)    Data collection. Avoid repeating the objectives of the study (pag. 3, line 133-134). It is important to note that the coordinators accepted the transcripts.

5)    Expand with a section on Rigor and Quality Criteria. Specify how concepts such as credibility, transferability and dependability were taken into account in the research. You could also refer to the evaluation through the COREQ qualitative design checklist (Tong et al., 2007).

Resultados

Congratulations to the authors for Figure 1.

In relation to section 3.3 (pag. 8, line 320) I would improve the wording in relation to the aspect of ethical compliance, it is not a barrier, it is a necessity for all research. Another thing is the process that must be followed, which is sometimes tedious. Later, in the discussion, it is better understood (pag.14-15, lines 548-558).

Discussion

Likewise, some more discussion elements could be incorporated on aspects such as the rural and cultural context that emerge in the results, they are very interesting. I suggest these aspects in the paragraph on pag.15, line 559-565.

Include a brief sections on Limitations..

Conclusions

I haven´t noticed the reference to Burton et al, in the conclusions. It seems like a justification of the importance of the research carried out.  

Author Response

Firstly congratulations to the authors for their work. I hope that my methodological comments will help you give more rigor to your study. 

Thank you for your comments. We hope that our correction following your suggestions made the paper clearer.

Abstract

Being a qualitative study, I suggest changing the verbs “identify y analyze” as they are verbs typically used in quantitative studies. Think of verbs like explore, investigate or others.

We changed it to “explore and understand”

Introduction.

The introduction is concrete and clear. I suggest the possibility of incorporating more bibliography in relation to, for example the "Chronic Disease Self-Management Program" (CDSMP) (pag 2, line 60-73) and the concepts of vulnerability (pag 2). The authors suggest a definition, but was this the adapted one for all countries, or just Spain? It may be nice to know how these criteria were agreed upon between countries.

Thank you for your comment. We added two references about effectiveness of the CDSMP (Griffiths et al., 2005; Turner et al., 2014). The definition was the one used during the project, we added a paragraph and reference with more details:

The final adopted definition reflect those used by the International Red Cross and the World Health Organization. It was agreed upon by all partners involved in the EFFICHRONIC project during one of the initial meetings, and defined as: “the diminished capacity of an individual or group to anticipate, cope with, resist and recover from the effect of natural or man-made hazard. Within the EFFICHRONIC project the term “vulnerable” included “people suffering from social exclusion and socio-economic hardship, as well as people under physical or psychological stress” [11] (p.14).

In this section, the entire EFFICHRONIC program must be presented, not in methodology. You will see that later on I suggest removing the section on Patient and public involvement.

Thank you for the suggestion, we modified accordingly.

Material and Methods

Separate the section into two: on the one hand design, and on the other the participants and context (pag. 3 line 116).

1)    Study design: A qualitative design is presented but the type needs to be specified. I suggest the authors look for methodological authors at the level of descriptive qualitative designs. And I suggest you read this article: doi:  10.1177/1744987119880234

We thank the reviewer for the suggestion. We followed Rendle et al 2019 : 10.1136/bmjopen-2019-030123 publication which described various qualitative studies, and added that it is an exploratory qualitative study, as there is not much literature on recruitment strategies, and we hope that our paper can be a contribution in this field.

2)    Participants and context: The truth is that the study is very interesting. When interviewing only 5 coordinators (these are the participants of this research) it is an important limitation (include in the limitations section). For this reason, it is essential in this section (or in results) to describe the study participants (coordinators). I suggest making a table (age, sex, basic training, advanced training, ..). Also describe the function of the coordinators within the program. It is important for the reader to be able to identify the profile and functions of the coordinators in order to understand the results. In this way it is better understood and it justifies why the study is carried out with so few participants. It would be necessary to briefly define the context in which the program has been carried out in the 5 countries.

We were unable to provide these kinds of details on the participants as being such a small sample, as noted, it would result in participants being recognizable. We did however specify more about their roles, we hope this helps.

We would also like to refer the reviewer to a paragraph in the introduction, p. 3 which presented some of these contextual differences as well, even though it does so in a general way to maintain anonymity:

In fact, EFFICHRONIC’s coordinators had to access the target population from very different professional backgrounds, different resources, and different organisational structures: university, research, hospital and governmental institutions. Recruitment has had its particularities in each of the five countries in the consortium due to different socio-cultural realities and social and health care models. The project leaders in each country have been in charge of tracing this intricate network and are, therefore, the people who know in detail how the process has been carried out.

3)    I do not fully understand the Patient and public involvement section (pag. 2, line 121); personally I would remove it. The first part (line 122-126) is included in the introduction and the second part (line 16-130) details the characteristics of the participants. I would include it in the Participants and context section. That is, the EFFICHRONIC program is presented in the introduction.

We modified this section following your suggestion, we hope it reads better now

4)    Data collection. Avoid repeating the objectives of the study (pag. 3, line 133-134). It is important to note that the coordinators accepted the transcripts.

We deleted the objective as suggested and added a line to specify about the transcript. 

5)    Expand with a section on Rigor and Quality Criteria. Specify how concepts such as credibility, transferability and dependability were taken into account in the research. You could also refer to the evaluation through the COREQ qualitative design checklist (Tong et al., 2007).

 We added a paragraph to describe with more details the analysis process as a collaborative process within the research team, we hope this helps the reader to understand the credibility and trustworthiness of the findings. We added a reference to cite, on issues of rigor in qualitative research

To enhance the credibility of the study, two researchers initially coded two of the transcripts separately and then compared the identified codes. This supported the main researcher to check whether important aspects were left out from her analysis because of unanticipated bias on the topic. The two researchers agreed on the identified codes and were then discussed together with the rest of the team who had also read the full transcripts. Moreover, in writing the results section, quotes from transcripts were used to support the description of the findings [14].

Resultados

Congratulations to the authors for Figure 1.

In relation to section 3.3 (pag. 8, line 320) I would improve the wording in relation to the aspect of ethical compliance, it is not a barrier, it is a necessity for all research. Another thing is the process that must be followed, which is sometimes tedious. Later, in the discussion, it is better understood (pag.14-15, lines 548-558).

We changed it to: The required processes to compliance with ethical requirements has slowed down project implementation in some countries

Discussion

Likewise, some more discussion elements could be incorporated on aspects such as the rural and cultural context that emerge in the results, they are very interesting. I suggest these aspects in the paragraph on pag.15, line 559-565.

The lines suggested refer to ethics committees, so we did not add details on this section. However, we did add a line in the introduction as to the specificity which the project had to deal with, which depended on each context.

Include a brief sections on Limitations..

We added it as follow:

Study strengths and limitations

The main limitation of this study is the limited number of participants interviewed. Nonetheless, this was due to the specificity of the way the EFFICHRONIC project was carried out and the precise topic which this nested study was centred on (perceived barriers and facilitators to recruitment). Nonetheless, it should be noted that the variability of the profiles of the interviewees and of the contexts of implementation has allowed for a cross-case comparison [17] and provided novel insights into the recruitment process at an international scale, which has been rarely studied. It is expected that despite the small number of participants, the present analysis still provides interesting findings on the barriers faced in recruitment and useful strategies that can be adapted to other settings.

Conclusions

I haven´t noticed the reference to Burton et al, in the conclusions. It seems like a justification of the importance of the research carried out.  

The reference is now corrected.

Reviewer 3 Report

Thank you for the possibility to review the manuscript entitled: “ Recruiting participants in vulnerable situations: a qualitative 2 evaluation of the recruitment process in the EFFICHRONIC 3 study”. The manuscript is very interesting since it aimed to identify and analyse the recruitment strategies implemented in the participating countries (Spain, UK, Netherlands, Italy, 20 and France)

 in the EFFICHRONIC project. The project reflects these principles and aims to reduce the bur-16 den of chronic diseases and increase the sustainability of the healthcare system through the implementation of an evidence-based chronic disease prevention and self-management programme.

The manuscript is very interesting, the quality of figure and tables are satisfactory, statistical methods are valid and adequately applied and methodology is well and clearly explained to allow replication studies, too. Please revised all the references and revised the section in order to be more compliant with the journal rules.

Author Response

Thank you for your comments. We revised all reference, following the complete guide of the journal, so we hope it now meets all the criteria.

Round 2

Reviewer 2 Report

Thank you for the review. Congratulations again with the article.

I like the methodological approach as qualitative exploratory. I just think it would be better to separate Materials and Methods into two different sections; a) Study Design (a qualitative explorative study design was adopted [12]) y b) Participants (the rest of the information about the participants). It is easy to separate.

 It was a pleasure reading your research.